# Numerical Investigation of Slope Stabilization Using Recycled Plastic Pins in Yazoo Clay

**Mohammad Sadik Khan \***![ID], **Masoud Nobahar and John Ivoke**

Department of Civil and Environmental Engineering, Jackson State University, 1400 J.R. Lynch Street, JSU BOX 17068, Jackson, MS 39217-0168, USA; j00816771@students.jsums.edu (M.N.); j00820558@students.jsums.edu (J.I.)

**\*** Correspondence: j00797693@jsums.edu; Tel.: +1-601-979-6376

**Abstract:** Geographically, at the center of Mississippi is a concentration of High Plastic Yazoo Clay Soil (HPYCS). Shallow landslides frequently occur in embankments constructed with HPYCS caused by rainfall-induced saturation of the embankment slope. The traditional methods are becoming expensive to repair the shallow slope failure. The use of Recycled Plastic Pins (RPPs) to stabilize shallow slope failures offers a significant cost and construction benefit and can be a useful remedial measure for these types of failures. The current study investigates the effectiveness of RPP in slopes constructed with HPYCS, using the Finite Element Method (FEM). The FEM analysis was conducted with the PLAXIS 2D software package. Three uniform and varied RPP spacings were investigated to reinforce 2–4H:1V slopes. Reinforced slope stability analyses were performed to investigate the applicability of RPP in HPYCS. The FEM analysis results indicated that RPP provides shear resistance for the sloping embankment constructed of HPYCS. Uniform spacing of RPP provides sufficient resistance that increases the Factor of Safety (FS) to 1.68 in 2H:1V slopes with deformation of RPP less than 15 mm. The uniform spacing and varied spacing combination of RPP increase the FS to 2.0 with the deformation of RPP less 7 mm.

**Keywords:** shallow slope failures; yazoo clay; recycled plastic pins; finite element method

## 1. Introduction

In Mississippi, the majority of highway slopes, embankments, and levees constructed of HPYCS experienced shallow slope failure repeatedly. These failures are characteristically caused by rainfall-induced saturation of the embankment slope and can be expensive to repair. Typically, continuous wetting of surficial soil layers causes pore water pressure to increase and soil strength to reduce [1,2]. This condition can be further intensified by moisture variation due to seasonal weather patterns that result in cyclic shrink and swell of HPYCS. The possible cause of shallow slope failure may be due to the infiltration of rainwater into the cracks, which generate a softened zone in the soil slope [3]. It is reported that slopes made of fine-grained soil experienced failures due to prolonged rainfall [2]. Over time, the Fully Softened Shear Strength (FSSS) merging with Perched Water Condition (PWC) induced rainfall is reported to be the principal cause of shallow slope failures [1,4]. Generally, the failure depth in embankments constructed of HPYCS is usually within 0.91–1.82 m and the slip surface along with the failure envelope remains parallel [1]. The role of unsaturated soil status in the prevention of natural slopes from failure is inevitable.

For shallow slope failures relatively less than 5 m, one of the beneficial remediation techniques is considered to be slope reinforcement. Typical in situ slope stabilization solutions have been included as soil nailing, drilled piers, micro piles, and recycled plastic pins. Compared to available slope stabilization remedial techniques, Recycled Plastic Pins (RPP) are considered to be a cost-effective solution [1,3,5]. RPPs are driven into the slope face perpendicularly to improve shear resistance and the FS along a slip plane. Recycled plastics and waste materials such as polymers, sawdust, and fly ash are the

principal source of fabrication for RPPs [6]. The benefits of the RPPs are included as being lightweight material and having no susceptibility to biological degradation. With the utilization of RPPS, less waste volume enters landfills and recycled plastic markets grow [1]. The typical composition of RPPs is included as high-density polyethylene (<70%), low density polyethylene (<10%), polystyrene (<10%), polypropylene (<7%), polyethylene-terephthalate (<5%), and additives (fly ash <5%, and etcetera) [7].

Reportedly, after successive rainfall, slopes experienced failures, particularly on fine classified soils [8–10]. The soil unsaturated zone can be considered as the impacting factor in terms of slope failure prevention. It is observed that a low suction value could hold a reasonable stabilization factor within the shallow soil deposit regarding slope morphology [11]. Moreover, due to the existence of soil cover, at the unsaturated state, the hydraulic conductivity remains low, which affects the soil not be fully saturated particularly at the time of heavy rainfall period. In most cases, with the presence of vegetation cover, evapotranspiration causes the topsoil part to lose its water content during the dry periods between rainfall events. Consequently, high intensity and short duration rainfall following a long duration rainfall can cause the soil to be fully saturated and trigger the slope to be slipped [4,11]. It should be noted that the aforementioned condition may not be the case for high plastic clay soils.

Khan et al. [12] presented a field study of a highway slope located along US 287 near St. Paul overpass in Midlothian, Texas. The slope was constructed with high plasticity clay (CH). A surficial movement was observed on the slope that included tension cracks in the shoulder. As a remediation technique, a total of three sections (designated as Reinforced Section RS 1, RS 2, and RS 3) were reinforced with RPP. Different spacing (i.e., 0.9, 1.52, and 1.82 m on center) of RPP was utilized in the RS 1.

On the other hand, RS 2 and RS 3 were stabilized using uniform 1.21 m on center spacing. The performance monitoring of the US 287 slope (Figure 1B) indicated that the settlement at RS 1 was 6 cm, whereas the crest settlements were 12 cm and 8 cm in RS 2 and RS 3, respectively. The settlement at RS 1 was low due to the smaller spacing of RPP (0.9 m) at the crest of the slope. The study summarized that closer spacing of RPP near the crest of the slope, where the tension crack initiates the movement of the slope would provide superior performance than the slope stabilized with the uniform spacing of RPP. During this study, the closer spacing of RPP near the crest and uniform spacing of RPP at the rest of the slope was utilized.

The current research study investigates the applicability of RPPs in slopes made of HPYCS numerically using the finite element method. Three slopes (2H:1V, 3H:1V, and 4H:1V) with uniform spacing of RPP and different RPP spacing were investigated. For a uniform spacing, 0.91, 1.21, and 1.52 m on center were selected based on several field investigations conducted in Missouri, Iowa, and Texas [13]. For the varied spacing, each slope was divided into two parts, designated as the top part (1/3 of the length near the crest) and the bottom part (the remaining 2/3 length of the slope). The technique included a constant 0.91 m spacing near the crest and 1.21, 1.52, and 1.82 m for the rest of the slope. The length of the RPP was 3.04 m. The Finite Element Method PLAXIS 2D software package was employed to investigate the effect of spacing on both slope FS and deformation on HPYCS.

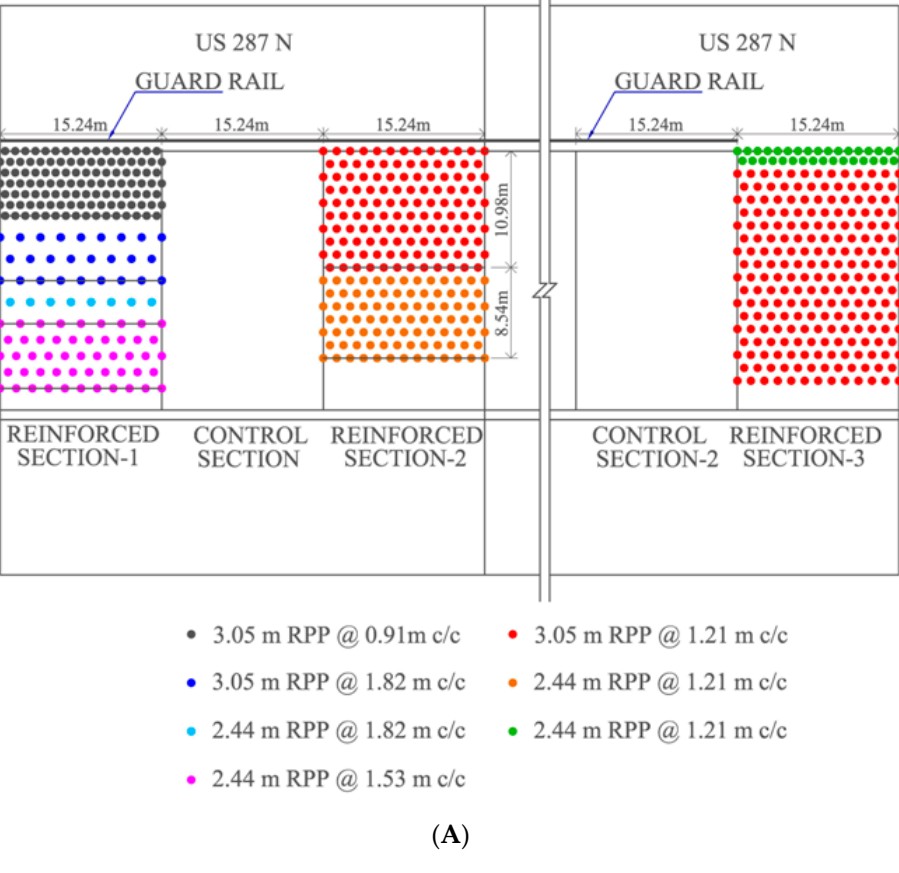

**(A)**

Total Settlement Profile at Crest

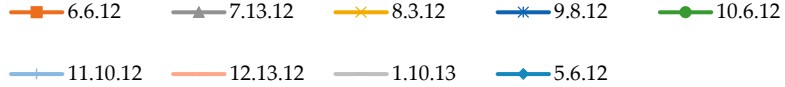

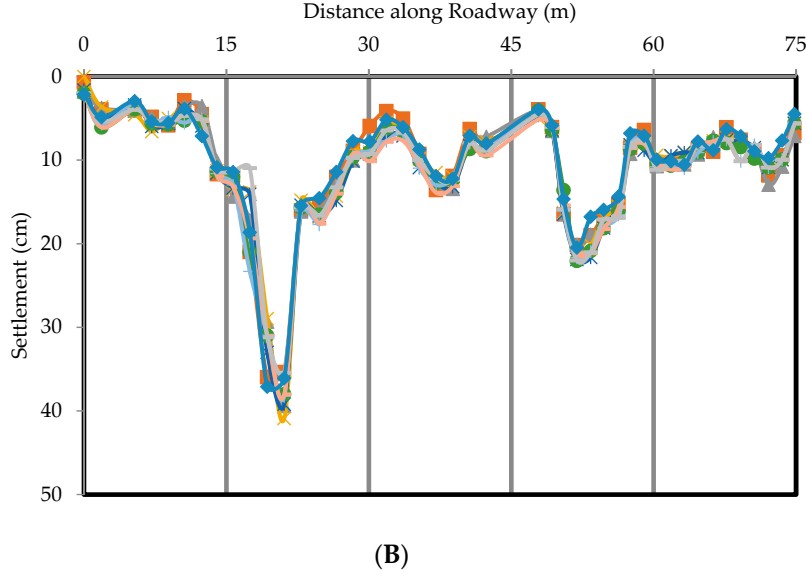

**(B)**

**Figure 1.** Performance of US 287 slope (**A**) the layout of RPP and (**B**) total settlement profile at the crest of the slope [11].

## 2. Site Location

The surficial slope failure is common on slopes constructed of HPYCS. A highway slope that was constructed using HPYCS having an early indication of slope failure was selected for this study. The Weathered HPYCS samples' physical properties (liquid limit, plasticity index, and gradation) were lab tested. The physical properties of the field soil samples were determined as the liquid limit of 108, and the plasticity index of 84, the percent passing #200 sieve of 90% [1]. The FSSS of the expansive soil was followed by the US Army Corps of Engineers' developed procedure. Based on the aforementioned procedure, the FSSS of the HPYCS resulted to be the cohesion of 5.22 kPa and friction angle of 18.7 degrees [1,2]. Khan et al. [1] investigated the effect of the rainfall on the matric suction and stability of the slope constructed on the HPYCS. The study indicates that the presence of a matric suction variation tends to affect stability. Successive rainfall infiltrates through the surface which tends to eliminate the matric suction and forms a temporary PWC in the slope. The combination of the fully-softened zone and rainfall induces temporary PWC that causes the shallow slope failure in slopes of expansive soil [2]. The current study considers the effect of the PWC along with the FSSS shear strength as a critical condition in FEM analysis, to investigate the slope stabilization scheme using RPP.

Khan et al. [2] investigated the HPYCS sample changes in shear strength test results. In this study, after different Wetting and Drying Cycles (WDCs) of zero, three, five, and seven repetitions, the cohesion and friction angles were calculated. It was observed that the soil cohesion strength reduced significantly as WDCs increased. The 18.44 kPa soil cohesive strength for zero WDCs was reduced to 4.31 kPa for the seventh WDC, which is a 77% drop in the strength. However, a drop was observed as well in friction angles, which was not significant. For instance, the 20.34-degree soil frictional angle for zero WDCs dropped to 18.42 degrees for the seventh WDC, which exhibited a 10% reduction. The study results were in good agreement with Wright and Zornberg's observations [14,15]. Zornberg and Skempton, in numerous case histories, reported that the shear strength was being mobilized along the soil slip surface, which can be correlated with the FSSS in addition to proposed recommendations for slope stability analysis in terms of shear strength before any landslide occurrences [15,16]. The mentioned slopes are known as first-time slides [15,16]. According to the test results, it can be concluded that the slope after the seventh WDC (4.31 kPa and frictional angle of 18.42 deg) experiences its first-time slide. Furthermore, according to the shear strength test results, Nobahar et al. [17] investigated the change in shear strength of HPYCS, and from the peak, FSSS, and the residual tests, the Mohr-Coulomb failure envelop was developed. It was observed that the peak shear strength test (c = 18.4 kPa and phi = 20.2 degrees) resulted in the highest shear strength, whereas the residual shear strength test (c = 5.45 kPa and phi = 12.8 degrees) resulted in the lowest shear strength. In the current study, it is worth mentioning that the adopted FSSS test results were c = 10.8 kPa and phi = 18.6 degrees.

## 3. Finite Element Modeling

The stability and deformation analysis of numerous geotechnical cases and applications were performed utilizing the two-dimensional Finite Element Method (FEM), the Plaxis 2D program. For stability analyses, the elastic-perfectly plastic, Mohr-Coulomb soil model, and 15 node-triangular elements were utilized in this modeling. The boundary condition was set to have Standard fixities at the sides of the slope [18]. The FS was evaluated according to the phi-C reduction analysis.

The highway slope is 9.15 m high with a slope ratio of 3H:1V. Limiting FS to 1.0, the slope experiences failure. The geometry of the slope is developed based on the existing borehole log and soil test results. The physical and engineering properties and characteristics of HPYCS were determined by lab testing [16,19]. Moreover, the FSSSs, as determined from the soil test results are utilized for the uppermost soil. Based on the stability analysis results, the FS of the 3H:1V slope was observed as 1.27. As a result, the slope is stable and requires no remediation. An additional back analysis was conducted after applying a PWC

at the topsoil layer, considering the effect of rainfall. As a result, the FS of the slope reduced to 1.18. The soil parameters for the FEM analysis are summarized in Table 1.

Three geometries of the slope with three different slope ratios of 2H:1V, 3H:1V, and 4H:1V were generated similar to the highway slope in consideration. The geometry of the slopes as shown in Figure 2, was further utilized in FEM analysis. Based on the stability analysis, the FS of the 2H:1V, 3H:1V, and 4H:1V slopes were 1.093, 1.277, and 1.62, respectively, before RPP installation. The slip surface for each of the slopes (at the dry condition and PWC) is presented in Figure 3.

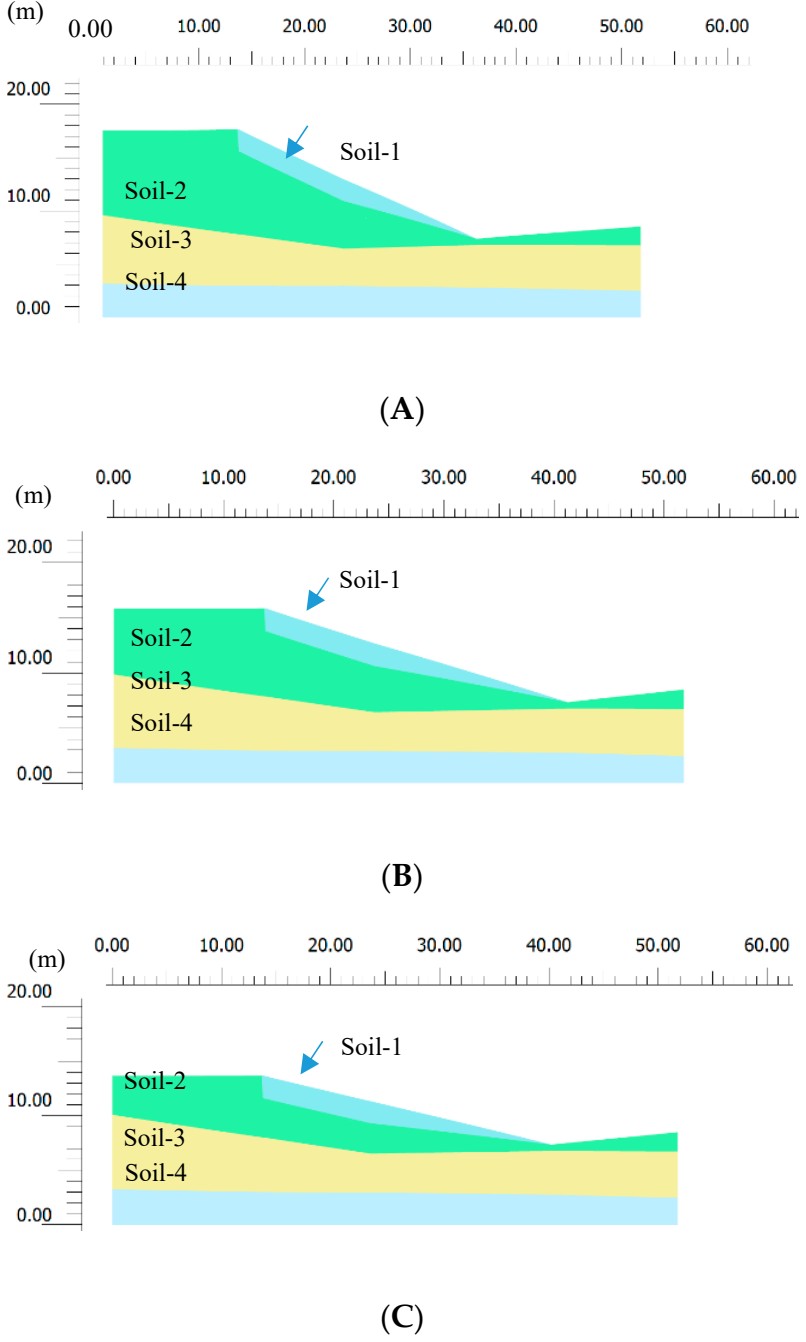

**Figure 2.** Slope geometries for (**A**) 2H:1V slope, (**B**) 3H:1V slope, and (**C**) 4H:1V slope.

**Table 1.** Parameters from the FE analysis.

| Soil Type | Friction angle $\varphi$ | Cohesion c | Unit Weight $\gamma$ | Elastic Modulus E | Poisson Ratio $\nu$ | RPP Parameters | | |
|---|---|---|---|---|---|---|---|---|
| - | ° | kN/m² | kN/m³ | kN/m² | - | | | |
| 1 | 18.6 | 10.8 | 21 | 4788 | 0.35 | EA | kN/m | 21.4E3 |
| 2 | 23 | 23.94 | 21 | 7183 | 0.30 | EI | kN m²/m | 1.310E6 |
| 3 | 25 | 47.89 | 22 | 9576 | 0.25 | d | m | 27.1 |
| 4 | 35 | 143.64 | 22 | 11,970 | 0.2 | | | |

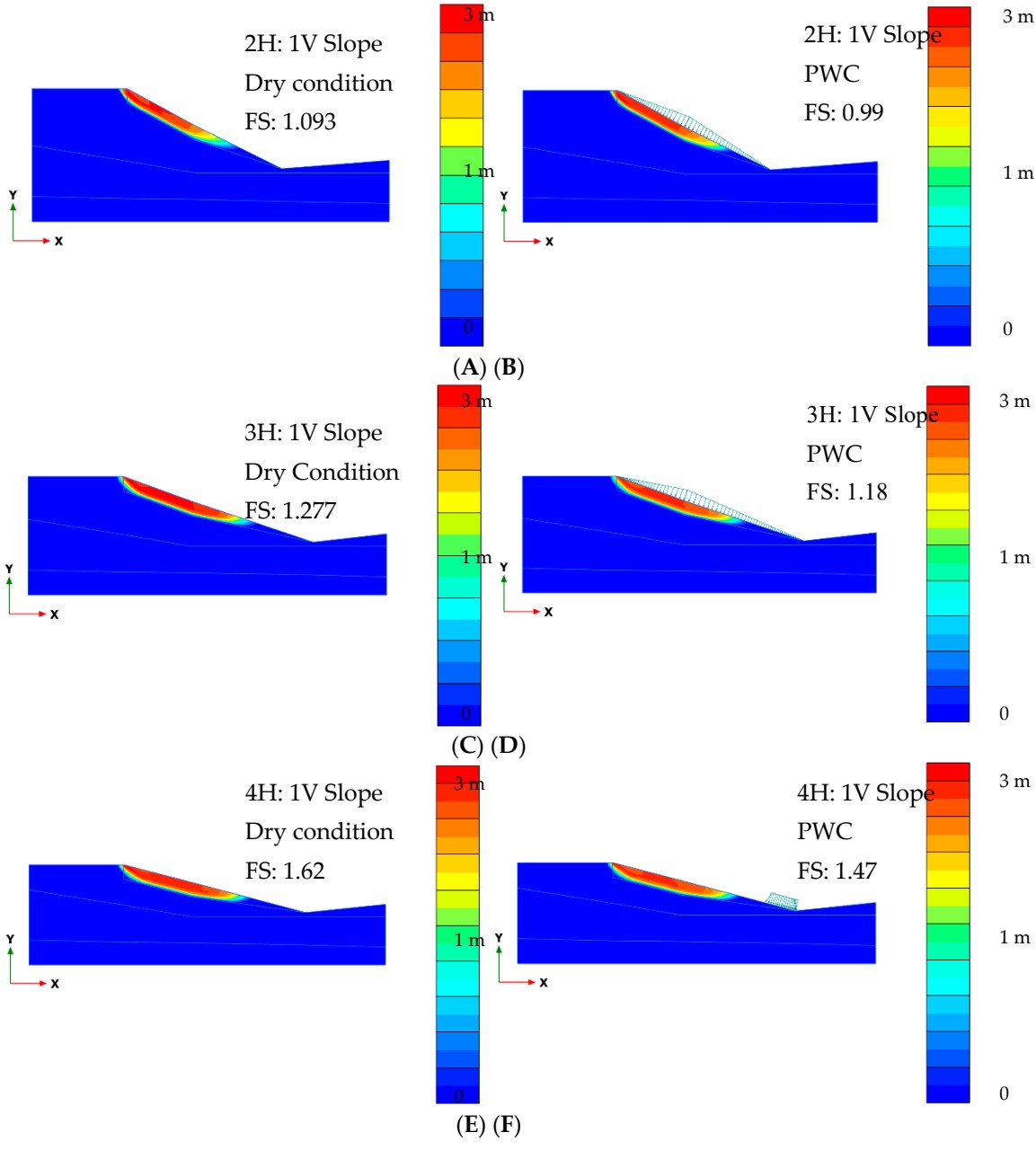

**Figure 3.** Stability analysis of the unreinforced slope: (**A**) 2H:1V slope (dry condition), (**B**) 2H:1V slope (PWC), (**C**) 3H:1V slope (dry condition), (**D**) 3H:1V slope (PWC), (**E**) 4H:1V slope (dry condition), and (**F**) 4H:1V slope (PWC).

To simulate the worst-case scenario, successive rainfall forms the temporary PWC that usually causes the shallow slope failure in Mississippi, a PWC considered in the topsoil [2]. The depth of the PWC was 2.13 m near the crest. To be able to similarly consider the pore water pressure inside the slope, the PWC was defined. The strength-reduction analysis was conducted to evaluate the FS of the slope with the PWC. The FEM analysis results indicate that the FS of the 2H:1V, 3H:1V, and 4H:1V slope reduced to 0.99, 1.18, and 1.47, respectively, when considering the PWC (Figure 3). This reduced FS is very close to failure for the 2H:1V and 3H:1V slopes. Thus, the unreinforced slope, considering the FSSS at the topsoil with the presence of PWC, was utilized to investigate the slope stabilization option of RPPs.

Khan et al. [20] studied the impact of WDC on the HPYCS shear strength variations based on several tests. During this study, tests were conducted on the samples of HPYCS collected from a highway slope site borehole in Jackson, Mississippi. The effect of WDC in a controlled environment in the laboratory was investigated on the HPYCS samples with 76.2 mm diameter and 25.4 mm height, which were subjected to three, five, and seven repeats of the WDC and then tested for the void ratio, microstructure, volumetric deformation, and shear strength. Khan et al. [20] reported the axial deformation of HPYCS under different WDCs. Furthermore, the fully-coupled flow and stability analysis with consideration of the soil unsaturated matric suction and moisture variation was also conducted. Both deformation and groundwater flow mixed equations are solved based on the coupled format of the time-dependent hydro-mechanical behavior of the soil simultaneously. With FEM numerical analysis, two rainfall periods of rainfall volume (RV) = 126.2 mm (2 h) and RV = 271.7 mm at three days were considered, and highlighting the effect of three, five, and seven WDCs in the topsoil, the FS resulted to be reduced from 1.7 to 1.2 (RV = 126.2 mm) and 1.68 to 1.02 (RV = 271.7 mm). Based on the numerical results, critical FS was observed, which was a coupled effect of the seventh WDC and rainfall volume impacting both the changes in shear strength and the saturation.

Finite element flow analysis conducted by Nobahar et al. [19] was utilized to evaluate the effect of rainfall on a slope constructed using HPYCS in Mississippi. During this study, representative HPYCS samples from a Mississippi highway slope were collected and tested. The physical and mechanical properties of the soil samples utilized from Nobahar et al. [16] study (liquid limit 108, plasticity index 84, dry unit weight 12.8 kN/m$^3$, specific gravity 2.68, vertical permeability value of $k_v$ = 0.034 cm/s). The direct shear test was utilized to determine the variation of shear strength of the HPYCS samples. Based on Nobahar et al. [19] study, the peak, FSSS, and the residual tests were determined through the Mohr-Coulomb failure envelop. The FSSS test results were c = 10.8 kPa and φ = 18.6 degrees. The flow analysis was extended to evaluate the effect of different intensities and durations of rainfall, based on the Partial Duration Series (PDS) based Intensity-Duration-Frequency (IDF) curve. The details of the flow analysis are presented in the following section. For the parametric study, a 100-year return period of the PDS-based IDF curve of precipitation based on the 2014 National Oceanic and Atmospheric Administration (NOAA) Atlas for Jackson, Mississippi was utilized. Different rainfall volumes (70.8 mm (2.7 in) and 271.7 mm (10.6 in)) with different rainfall durations (2 h to 3 days) were selected based on the area of the IDF curve and NOAA database.

To assess the coupled flow-deformation behavior of the soil model during the precipitation, various total rainfall volumes (70.8 mm (2.7 in) to 312.4 mm (12.2 in)) were implemented. Supposing the duration of precipitation lasted 30 min, 60 min, 2 h, 6 h, 12 h, 1 day, and 3 days, for each precipitation intensity, the flow within the topsoil of the slope was evaluated. The hydraulic model was selected to be the Van Genuchten model. The HPYCS Water Retention Curve (SWRC) curve was developed by Nobahar et al. [17] in Mississippi. During the numerical analysis, the Van Genuchten fitting parameters of the HPYCS were employed.

## 4. RPP Configuration for Slope Stabilization Option

The current study investigates the performance and safety of a reinforced slope on HPYCS using RPP. As a part of the study, each slope ratio is divided into two parts designated as the top part (1/3 of the length near the crest) and the bottom part (the remaining 2/3 of the length of the slope).

The top part of each slope ratio is reinforced with 0.91 m RPP spacing to provide additional resistance against deformation, while the rest of the slope is treated with a higher spacing of RPP. The RPP is modeled as the plate element. The properties of RPP are also summarized in Table 3. A total of six different RPP spacings were considered for each slope ratio, Khan et al. [11], which was designated as configurations A, B, C, D, E, and F. RPP configurations A, B, and C were modeled with uniform RPP spacing of 0.91, 1.21, and 1.52 m, respectively. Varied RPP spacings were utilized in configurations D, E, and F, where at the top part RPPs are kept at constant spacings of 0.91, and 1.21, 1.52, and 1.83 m. RPP spacing is considered at the bottom part of each slope. The details of the study are summarized in Table 2 and Figure 4.

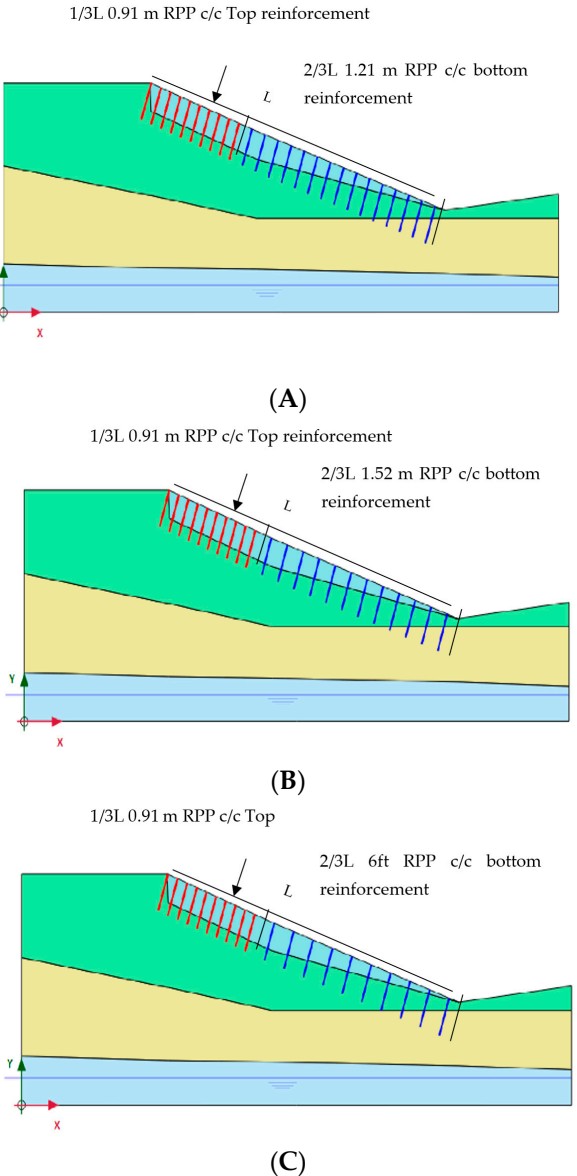

**Figure 4.** RPPs layout at the top and bottom part of the slope: (**A**) configuration D, (**B**) configuration E, and (**C**) configuration F.

**Table 2.** Numerical modeling matrix.

| Slope Ratio | RPP Configuration | RPP Spacing for Full Length of Slope | RPP Spacing at 1/3 of the Length of Slope near the Crest (Top Part) | RPP Spacing at 2/3 of the Length of the Slope near the Crest (Bottom Part) |
|---|---|---|---|---|
| 2H:1V, 3H:1V, and 4H:1V | A | 0.91 m | - | - |
| | B | 1.22 m | - | - |
| | C | 1.52 m | - | - |
| | D | - | 0.91 m | 1.22 m |
| | E | - | 0.91 m | 1.52 m |
| | F | - | 0.91 m | 1.83 m |

## 5. Slope Stability Analysis of the Reinforced Slope

Employing the strength-reduction technique, the slope stability factor was evaluated. Various RPP spacings with a change in slope stability factor are presented in Figure 5. The FS of each RPP configuration is also summarized in Table 3.

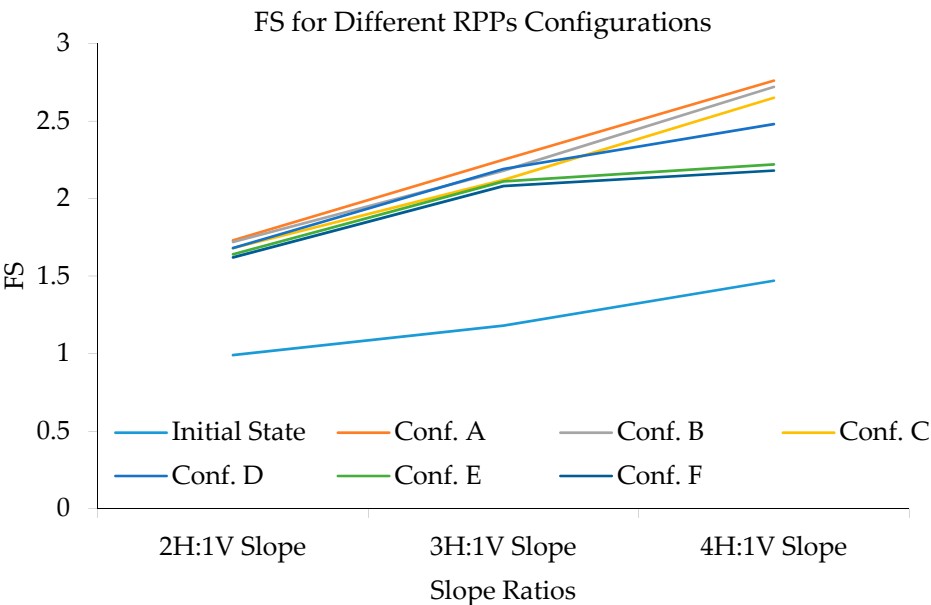

**Figure 5.** Variations in the FS for different RPP configurations for the three slope ratios of 2H:1V, 3H:1V, and 4H:1V.

**Table 3.** FS values for different RPP spacings in the 2:1, 3:1, and 4:1 slopes.

| Slope Ratio | Initial FS | RPP Configuration | | | | | |
|---|---|---|---|---|---|---|---|
| | | **A** | **B** | **C** | **D** | **E** | **F** |
| | | 0.91 m RPP Spacing | 1.21 m RPP Spacing | 1.52 m RPP Spacing | 0.91 and 1.21 m RPP Spacing | 0.91 and 1.52 m RPP Spacing | 0.91 and 1.82 m RPP Spacing |
| 2H:1V Slope | 0.99 | 1.73 | 1.72 | 1.68 | 1.68 | 1.64 | 1.62 |
| 3H:1V Slope | 1.18 | 2.25 | 2.18 | 2.12 | 2.19 | 2.11 | 2.08 |
| 4H:1V Slope | 1.47 | 2.76 | 2.72 | 2.65 | 2.48 | 2.22 | 2.18 |

Based on the FEM analysis results, the 0.91 m spacing (configuration A) provides the highest FS of the 2H:1V slope. The FS of the unreinforced 2H:1V slope was 0.99, whereas including the 3.04 m long RPPs, which were placed at 0.91 m on center increased the FS to 1.73. Moreover, the FS with 1.21 m spacing (configuration B) was close to the FS of

0.91 m spacing. With 1.52 m spacing (configuration C), a slight drop in the FS was observed, with a value of 1.68. The FS of the configuration D (0.91 m spacing near the crest and 1.21 m spacing near the toe) had the same value of the FS as configuration C. Besides, the FS of configuration E (0.91 m spacing near the top and 1.52 m spacing near the toe) and configuration F (0.91 m spacing near top and 1.83 m spacing near the toe) had an FS of 1.64 and 1.62, respectively.

The increment of the FS of the 3H:1V and 4H:1V slopes followed a similar trend with different RPP configurations as the 2H:1V slope. Based on Figure 5, the FS of the slope increased to 2.12 and 2.64 for the 3H:1V and 4H:1V slopes with 0.91 m spacing all over the slope. The lowest FS of the reinforced 3H: 1V and 4H: 1V slope was calculated to be as 2.08 and 2.18 respectively, with RPP configuration F (0.91 m spacing near crest and 1.83 m spacing near the toe of the slope).

As presented in Table 3, with the PWC, the 2H:1V and 3H:1V had a marginal FS of 0.99 and 1.18, respectively. With the inclusion of −21 of the RPPs at different configurations, the FS of the reinforced slope increased to a range between 1.62 and 2.08, higher than 1.5 which is required by several state and federal agencies for slope repair. Regarding safety, all the RPP configurations provide enough support to make the reinforced slope stable.

## 6. Deformation at the Crest

The plastic analysis was performed in Plaxis 2D for deformation analysis. An initial condition of the slope was generated using a gravity loading condition which is recommended for non-level ground, such as slopes. During gravity on analysis, the vertical stress is calculated based on the weight of unstressed mesh, and then horizontal stresses are changed to be equal to k0 (earth pressure coefficient at rest) times the calculated vertical stress. Later, the plastic analysis was conducted by activating the RPPs in the soil model.

The maximum lateral deformation of 3.04 m long RPP at the crest of the slope with different RPPs configurations at the top section is presented in Figure 6. The horizontal displacement plots showed that the RPP had rotational movement. Moreover, the RPP experienced a translational movement near the base, which implied that it is not fixed at the base. The movement of the RPP had similar movement to that of the short pile, which does not have enough fixities in the slope. It should be noted that the depth of soft soil in the FEM analysis was 2.13 m. As a result, the 3.04 m long RPP had only 1/3 of its length into the stiffer base. However, the deformation of the slope is negligible. During this study, only the deformation of the slope at the crest (from the top section) is presented. The deformation of the bottom section (at middle and toe) of the slope was observed to be less compared to the displacement at the crest. Therefore, no deformation result from the bottom section of the slope is presented here.

Based on the plastic analysis results, the uniform 0.91 m (configuration A), 1.21 m (configuration B), 1.52 m (configuration C), and the spacing of RPPs provided enough resistance at the crest of the 2H:1V slope. As a result, the horizontal movement of the RPPs is stabilized within a range of 12 mm. However, for the varied spacing of RPPs, 0.91 m/1.21 m (configuration D), 0.91 m/1.52 m (configuration E), and 0.91 m/1.83 m (configuration F) have relatively large deformations at the top of the RPPs within a range of 225 mm or greater. It should be noted that the configurations D, E, and F had a high FS within a range of 1.62 and 1.68. Therefore, the configurations D, E, and F are not recommended in the 2H:1V slope.

The uniform RPPs spacings of 0.91 m (configuration A), 1.21 m (configuration B), 1.52 m (configuration C), different spacings of 0.91 m/1.21 m (configuration D), 0.91 m/1.52 m (configuration E), and 0.91 m/1.83 m (configuration F) had a similar movement trends at the crest of the 3H:1V and 4H:1V slopes. Based on Figure 6, the maximum horizontal displacement at the RPP on top was observed as 7 mm, which is almost negligible. Also, all the configurations had a high FS with 3H:1V and 4H:1V slopes. Since the varied spacing with 0.91 m at the top part and 1.21 m to 1.83 m at the bottom part require less RPP to reinforce the slope, the cost of the slope repair would be less. Therefore, the configurations D, E, and F would be suitable for 3H:1V and 4H:1V slopes.

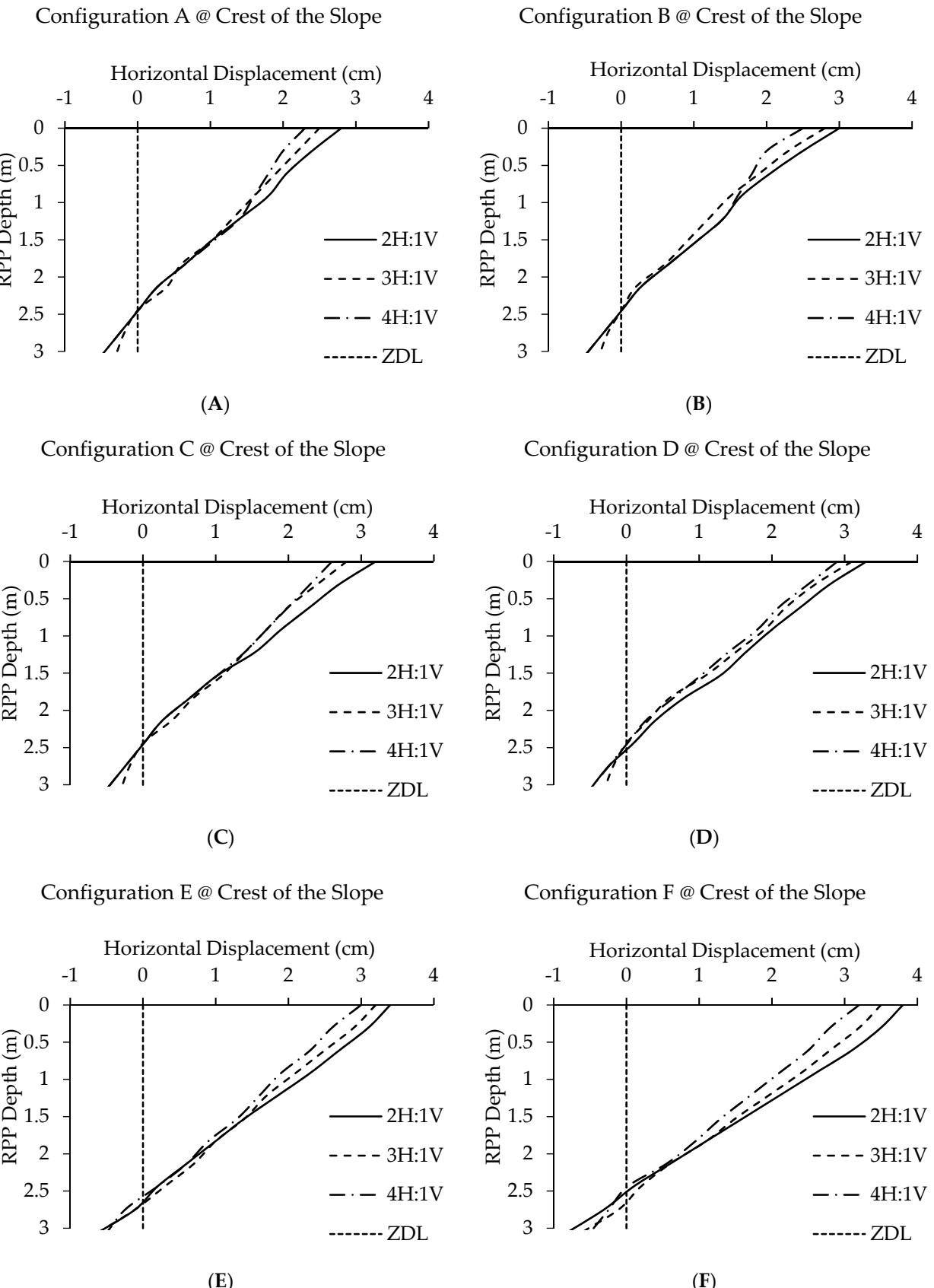

**Figure 6.** Horizontal displacement profile of RPPs at the crest of the highway slopes: (**A**) configuration A, (**B**) configuration B, (**C**) configuration C, (**D**) configuration D, (**E**) configuration E, and (**F**) configuration F (Zero Displacement Line (ZDL)).

Moreover, the movement of the RPP had similar movement to the short pile, which does not have enough fixities in the slope. The horizontal displacement plots showed that the RPP had rotational movement and experienced a translational movement near the base, which implied that it is not fixed at the base. In this regard, it is expected that the RPPs would have no sequential pattern of displacement with slope failure and configurations. Also, considering six RPP configurations for three slope geometries, the results showed that RPP displacement at the crest of the slope where the maximum slope failure occurs differs.

## 7. Conclusions

Shallow slope failure is typical and recurring in many highway slopes in Mississippi constructed with HPYCS. The RPP could be a beneficial method to stabilize such shallow slope failures. The current study investigates different RPP configurations at 2H:1V, 3H:1V, and 4H:1V slopes numerically. Stability and deformation analysis of the reinforced slopes have been performed to assess the applicability of RPP in HPYCS. Based on the 2D FEM numerical analysis, it can be concluded that:

- RPPs provided resistance at the slope and increased the FS. Since the 3.04 m-long RPP has been investigated in the current study, all the RPPs acted as a short pile to increase the stability of the slope.
- The uniform spacing of RPPs (configuration A, B, and C) resulted in a high FS and significantly less deformation near the crest of the 2H:1V slope. The varied spacing of RPPs (configuration D, E, and F) resulted in a high FS. However, it has significant deformation near the crest of the slope. Thus, the uniform configuration is most suitable for a 2H:1V slope.
- The spacing of RPPs at the top section had a significant effect on the deformation of the slope for 3H:1V and 4H:1V slopes, when the failure of the slope initiated from the crest. With closer RPP spacing at the top section, the deformation of the slope was low. As a result, the effect of RPPs spacings at the bottom section was not significant for the performance. However, with an increase in RPPs spacings at the bottom section, the FS of the slope decreased. The varied spacing of RPPs (configuration D, E, and F) have similar safety and performance as the uniform spacing of RPP (configuration A, B, and C). Since the varied spacing requires a smaller number of RPPs, it could be cost-prohibitive and thus desirable only for 3H:1V and 4H:1V slopes.

**Author Contributions:** Conceptualization, M.S.K.; Data curation, M.N.; Formal analysis, J.I.; Funding acquisition, M.S.K.; Investigation, J.I.; Methodology, J.I.; Project administration, M.S.K.; Resources, M.S.K.; Software, M.N.; Validation, M.N.; Writing—Original draft, J.I.; Writing—Review & editing, M.S.K. and M.N. All authors have read and agreed to the published version of the manuscript.

**Funding:** This study is supported by the U.S. Department of Transportation under Grant Award Number DTRT13-G-UTC50. The work was conducted through the Maritime Transportation Research and Education Center at the University of Arkansas.

**Data Availability Statement:** The data presented in this study are available on request from the corresponding author.

**Acknowledgments:** The authors would like to thank and acknowledge the U.S. Department of Transportation. The findings, conclusions, and recommendations expressed in this material are those of the authors and, necessarily, it does not reflect the viewpoints of the MDOT.

**Conflicts of Interest:** The authors declare no conflict of interest.

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
