# Peer review of "Numerical Investigation of Slope Stabilization Using Recycled Plastic Pins in Yazoo Clay"

_infrastructures, doi:10.3390/infrastructures6030047_

Round 1
Reviewer 1 Report
The object of the paper is interesting for the soil slope failure analysis. The manuscript is well organized. I suggest the paper be accepted for publication.
Author Response
The authors sincerely appreciated the positive feedback of the reviewer

Reviewer 2 Report
The manuscript investigates the effectiveness of Recycled Plastic Pin to repair the shallow slope failure in slopes constructed with High Plastic Yazoo Clay, through a series of numerical analyses based on the Finite Element Method. The FEM are implemented using the commercial software Plaxis 2D. The topic is very interesting, and the novelty of the study is clear and well explained in the manuscript. In particular, the use of RPP represents a good solution to increase the factor safety of the slope considering also the costs in relation to other types of interventions. The numerical analyses are accurately conducted and the results obtained are clearly justified in the manuscript. For these reasons, it is opinion of this reviewer that the manuscript should be considered for publication in Infrastructures Journal, after the following minor/corrections Improvements:
- The manuscript is well-written. Only some small corrections are suggested:
In line 73 delete the second parenthesis in (RS) 1.
make Figure 1a more readable.
in the sentence “Recycled 49 plastics and waste materials such as polymers, sawdust, and fly ash are the principal 50 source of fabrication for RPPs [1]” consider the citation Longarini N., Crespi P., Zucca M., Giordano N. and Silvestro G. “The advantages of fly ash use in concrete structures”, Inzynieria Mineralna, 2014, 15(2), pp. 141–145
align the columns in Table 1
in the sentence “The boundary 1 condition was set to have Standard fixities at the sides of the slope” consider the citation Zucca M., Valente M., On the limitations of decoupled approach for the seismic behaviour evaluation of shallow multi-propped underground structures embedded in granular soils. Engineering Structures, 2020, 211, 110497
format the third column of the Table 2
Author Response
The authors sincerely appreciated the review comments of the reviewer. All the comments are addressed. Please check the attached file for specific responses.

This manuscript is a resubmission of an earlier submission. The following is a list of the peer review reports and author responses from that submission.